# Enabling Coarse X-ray Fluorescence Imaging Scans with Enlarged Synchrotron Beam by Means of Mosaic Crystal Defocusing Optics

**DOI:** 10.3390/ijms23094673

**Published:** 2022-04-23

**Authors:** Jonas Baumann, Christian Körnig, Theresa Staufer, Christopher Schlesiger, Oliver Schmutzler, Florian Grüner, Wolfgang Malzer, Birgit Kanngießer

**Affiliations:** 1Analytical X-ray Physics, Technical University of Berlin, Hardenbergstr. 36, 10623 Berlin, Germany; christopher.schlesiger@tu-berlin.de (C.S.); wolfgang.malzer@tu-berlin.de (W.M.); birgit.kanngiesser@tu-berlin.de (B.K.); 2Fachbereich Physik, Universitö at Hamburg and Center for Free-Electron Laser Science (CFEL), Luruper Chaussee 149, 22761 Hamburg, Germany; ckoernig@mail.desy.de (C.K.); theresa.staufer@desy.de (T.S.); oliver.schmutzler@desy.de (O.S.); florian.gruener@desy.de (F.G.)

**Keywords:** X-ray fluorescence imaging, synchrotron beam, HOPG/HAPG optics, coarse scan

## Abstract

Trace elements, functionalized nanoparticles and labeled entities can be localized with sub-mm spatial resolution by X-ray fluorescence imaging (XFI). Here, small animals are raster scanned with a pencil-like synchrotron beam of high energy and low divergence and the X-ray fluorescence is recorded with an energy-dispersive detector. The ability to first perform coarse scans to identify regions of interest, followed by a close-up with a sub-mm X-ray beam is desirable, because overall measurement time and X-ray dose absorbed by the (biological) specimen can thus be minimized. However, the size of X-ray beams at synchrotron beamlines is usually strongly dependent on the actual beamline setup and can only be adapted within specific pre-defined limits. Especially, large synchrotron beams are non-trivial to generate. Here, we present the concept of graphite-based, convex reflection optics for the one-dimensional enlargement of a 1 mm wide synchrotron beam by a factor of 5 to 10 within a 1 m distance. Four different optics are tested and characterized and their reflection properties compared to ray tracing simulations. The general shape and size of the measured reflection profiles agree with expectations. Enhancements with respect to homogeneity and efficiency can be expected with improved optics manufacturing. A mouse phantom is used for a proof-of-principle XFI experiment demonstrating the applicability of coarse and fine scans with the suggested optics design.

## 1. Introduction

X-ray fluorescence imaging (XFI) allows for quantitative mass determination of trace elements and labeled entities by detecting the element-specific characteristic X-rays upon excitation by a primary X-ray beam. In recent years, the use of XFI has gained traction in in situ and in vivo pre-clinical studies in small animals [1,2,3,4,5]. Potential applications range from tumor detection based on the uptake of functionalized nanoparticles to drug tracking by embedding fluorescence markers into the substance or immune response studies via labeled cells [6]. Optical fluorescence or bioluminescence imaging (BLI) allow for similar applications with the advantage of short acquisition times and no radiation damage, but suffer from strong attenuation losses, which limit the possible imaging depth to around 10 mm. Furthermore, scattering of the fluorescence inside the tissue limits the spatial resolution to a few millimeters when not emitted close to the surface [7]. By contrast, due to the greater penetration depth of X-rays, XFI allows for probing deep into tissue with only minimal deterioration of resolution. By tuning the marker element and primary X-ray energy to the sample size, XFI might even be applied to human-sized objects [4].

However, standard XFI measurement schemes rely on scanning the probe with a pin beam, which is an inherently slow method. To reduce the total scan time and radiation dose, a coarse and low statistics scan can be used to determine the region of interest, which can then be scanned again with superior spatial resolution and higher statistics. This two-step procedure is needed, as it is often not known a priori where the accumulation of XFI-markers is located. For the highest possible sensitivity, synchrotrons are needed to minimize background contributions from multi-order Compton scattering in the fluorescence energy region. On the downside, many synchrotron (SR) beamlines only provide a small incident beam of up to 1 × 1 mm^2^. In this case, coarse scans might be performed by a one-dimensional enlarged SR beam (sheet beam) and a continuous movement of the probe in the perpendicular direction.

One possibility to create a sheet beam is to exploit the larger divergence of bending magnet radiation in contrast to wiggler or undulator radiation and use large collimator slits in the respective direction. For example, the BAMline at the synchrotron radiation facility BESSY II in Berlin offers a horizontal beam width of up to 10 mm [8]. A further possibility is the application of focusing optics [9] to artificially increase the beam divergence. However, optics based on total reflection offer only small reflection angles at energies above 50 keV, leading to either large and expensive optics or small effective areas that can be used from the original beam due to the limited projected size of the optics. Additionally, the limited access to these specific beamlines further reduces the availability of the method. Zhao and co-workers used another approach based on Bragg crystals to enlarge a synchrotron beam in one dimension [10]. Here, the difference of incidence angle with respect to the surface and Bragg angle of an asymmetric cut Si(220) crystal is exploited. We also suggest the usage of reflective optics using Bragg reflection. Since the bending and shaping of ideal crystals is complicated and limited with respect to the radius of curvature, we use graphite-based mosaic crystals (highly oriented pyrolytic graphite (HOPG) and highly annealed pyrolytic graphite (HAPG)) applied to a glass substrate. Indeed, the crystals have the shape of thin sheets that can adhere on even strongly bent surfaces, making them attractive for optics design. In this work we present the concept of the defocusing optics, compare homogeneity and reflectivity characterization measurements with ray tracing simulations, and offer a first proof-of-principle XFI application on a mouse phantom.

## 2. Materials and Methods

### 2.1. Concept of Diverging Optics

With respect to geometric optics, defocusing or diverging of a parallel beam can be achieved by reflection of the beam at a bent surface. In the current case, the X-rays used for X-ray fluorescence imaging have an energy of E=53.04 keV ($\lambda_{E}=0.0349$ nm). To achieve high reflection efficiencies, Bragg reflection can be used. As mentioned in the introduction, HOPG and HAPG crystals allow for a flexible optics design due to the possibility to apply those materials on surfaces with small radii of curvature. With a lattice spacing of $d_{C}=0.3354$ nm for graphite, this corresponds to a Bragg’s angle of $\alpha_{B}=\arcsin(n\lambda_{E}/2d_{C})=$ 2.00° for n=1 the first order of reflection. Thus, the incidence angle on the optics should be 2.00° with a tolerance defined by the mosaic spread γ of the mosaic crystal, which can realistically reach, e.g., γ= 0.1° [11]. In the easiest case, a cone-shaped optics with the rotational axis being the *z* axis and with an opening angle of 2.00° fulfills this requirement for any beam parallel to the *z* axis (see Figure 1). From rotation symmetry follows that the shape of reflections from X-rays traveling at a radius *r* from the *z* axis is again a cone with an opening angle of twice the incident angle. If the incident X-ray beam has a finite width, it will hit the cone-shaped optics within a specific arc. Since the angle of the arc is constant on the reflection cone, the length of the arc will increase with increasing *z*.

Consider an X-ray beam traveling at $x=x^{\prime }$ and y=0 along the *z*-direction. Let the length of the arc of the cone-shaped optics that is hit by the incoming X-rays at radius $x^{\prime }$ be $l_{o}$ and the cone tip be in the center of origin. Then, the cone tip of the reflection cone is at $z_{cr}=\frac{x^{\prime }}{\tan(\alpha_{B})}-\frac{x^{\prime }}{\tan(2\alpha_{B})}$. Furthermore, the angle $\phi_{o}$ of the arc of length $l_{o}$ is $\phi_{o}=\frac{l_{o}}{x^{\prime }}$. At a distance $z^{\prime \prime }$, the radius of the reflection cone is $R_{r}=\left(z^{\prime \prime }-z_{cr}\right)\tan\left(2\alpha_{B}\right)$ and thus the length of the arc at $z^{\prime \prime }$, which in first approximation equals the width of the enlarged spot, which is:(1)$$\begin{array}{ccc} l_{r} & = & R_{r}\times \phi_{o} \end{array}$$(2)$$\begin{array}{ccc} & = & \left(z^{\prime \prime }-\left(\frac{x^{\prime }}{\tan(\alpha_{B})}-\frac{x^{\prime }}{\tan(2\alpha_{B})}\right)\right)\tan\left(2\alpha_{B}\right)\times \frac{l_{o}}{x^{\prime }} \end{array}$$

With $z^{\prime \prime }=1.185$ m, $\alpha_{B}=$ 2°, $x^{\prime }=12.9$ mm and $l_{o}=1$ mm, $l_{r}$ results in 5.4 mm at a distance of $z^{\prime \prime }-z_{cr}=1$ m away from the optics’ hit position. Indeed, to keep the distance short between the optics and enlarged spot, e.g., in order to reduce attenuation and scattering in the air and minimize the spot displacement, the radius of the curvature at the hit position of the cone-shaped optics ($x^{\prime }$) must be kept small. This can easily be realized with the flexibility of HOPG or HAPG crystals.

Due to small Bragg’s angles and finite mosaic spread, a tilted cylinder can be used to approximate the cone-shaped optics with negligible performance loss (Appendix A). Since cylindrically shaped optics are more readily available, we decided to use those for the investigations in this work.

### 2.2. Ray Tracing Simulations

Since the first geometric considerations for the diverging optics are very simplified, ray tracing calculations are performed to include a more precise description of the setup geometry and the reflection behavior of the mosaic crystal. Several computer codes exist to perform the task, e.g., SHADOW [12] or mmpxrt [13], both allowing to include the specific mosaic crystal behavior in the simulations. For this study, the reflection model for mosaic crystals and the ray tracing presented by Schlesiger et al. have been used [14]. In the ray tracing, the X-ray source is defined by a number of photons (typically 10^5^ to 10^7^), each with a position and direction vector as well as an energy. From the optics’ shape and orientation, the hit position and incidence angle of each photon on the optics’ surface is calculated. Then, the reflection properties of the HOPG or HAPG crystal are used to randomly sample a new direction for each photon, to correct the position for penetration effects into the crystal and to give each photon a weight factor, which accounts for each individual reflection probability. The intensity of the reflected photons can then be displayed on an arbitrary plane in the coordinate system.

### 2.3. Diverging Optics

Four different optics have been purchased from Optigraph GmbH (Table 1). They consist of convex, cylindrical substrates (Thorlabs) and adhesively applied HAPG and HOPG mosaic crystals.

### 2.4. Experimental Setup

Characterization of the diverging optics and X-ray fluorescence imaging proof-of-principle measurements were performed at the P21.1 broad band diffraction beamline at the synchrotron radiation facility PETRA III in Hamburg. The single-bounce monochromator was set to 53.04 keV. For adjusting the beam properties, several 0.1 mm thick thallium absorber plates can be introduced to reduce the beam intensity and a pair of slits can be applied to decrease the beam size from its original 1 × 1 mm^2^ down to 0.1 × 0.1 mm^2^.

In the experimental hutch the heavy load diffractometer (Huber, two rotational and three translational axes) and two 2D X-ray detectors have been used. The first detector (“X-ray Eye”) is an in-house development at the beamline and based on a scintillator crystal (YAG:Ce, 200 μm thickness), a lens system and a CCD camera. Its design is similar to detectors as in [15] and provides a resolution of about 9.9 × 9.9 ${\mu m}^{2}$. The second detector (“PE”) is a Si-based digital X-ray flat panel detector (Perkin Elmer XRD1621, 2048 × 2048 pixel with a size of 200×200 μm^2^), rendering a much larger detection area and better sensitivity but lower spatial resolution.

The X-ray beam was aligned to the pivotal point of the diffractometer on which the optics was placed (see Figure 1, right) to divert the incoming beam horizontally. The interchangeable detectors were positioned downstream at a distance of 1500 mm if not stated otherwise. The XFI stage was placed at about 1 m downstream, between the diverging optics and the detector, on an x−ϕ stage, allowing it to be aligned either to the direct or reflected beam or moved out of the beam during optics characterization. On this stage, a 3D-printed mouse phantom was mounted on the *x*–*y* translation axis and a silicon drift detector (SDD) with 50 mm^2^ collimated area and 0.5 mm thickness (X-123FASTSDD, Amptec Inc., Bedford, MA, USA) was used to obtain fluorescence spectra. The detector was placed at about 150° and 80 mm distance to the phantom.

As materials for the phantom, gypsum was chosen as bone-equivalent and Formlabs Clear Resin as tissue-equivalent material, providing realistic attenuation properties [16,17] as expected for small animal XFI measurements. The model contained cavities for different organs, which were filled with varying concentrations of palladium diluted in agarose to mimic uptake in organs. The phantom was constructed using the digital mouse atlas “digimouse” dataset [18].

## 3. Results and Discussion

### 3.1. Characterization of Optics

The non-diverged X-ray beam was imaged with the “PE” detector at two positions along the beam path, once at the default position of 1500 mm and once at 2036 mm distance from the diffractometer’s pivotal point. Additionally, an image of the non-diverged X-ray beam was recorded at the default detector position with the “X-ray Eye”. The measurements (Appendix A) show that the spot is of rather quadratic shape with a width of 1 mm and a slight increase of intensity towards the left center. From the“PE” measurements it can be seen that no increase in spot size within the 435 mm increase of distance is detectable, leading to the conclusion that the divergence is well below 0.5 mrad (0.03°).

The data were used to define the source properties in the ray tracing simulations. The positions of the source photons were sampled from a symmetric 2D Gaussian distribution at z=−23 m with a full width at half maximum (FWHM) of 1.1 mm. The directions were defined by straight lines to another Gaussian distributed random set of positions at z=−1.5 m, the FWHM being 0.92 and 0.9 mm for the *x* and *y* directions, respectively. Then, all photons that did not pass a quadratic aperture at z=−1.5 m of width 0.98 and 0.95 mm for the *x* and *y* directions, respectively, were removed. The energy of each photon was sampled from a Gaussian distribution centered at 53.04 keV and with a full width at half maximum (FWHM) of 53.04 eV to account for the beamline energy resolution.

Evaluating the reflected and direct intensity on the“PE” detector, the optics were aligned by means of successive rotational and translational scans with the diffractometer, such that the X-ray beam hit the center surface position of the optics with an incidence angle equal to Bragg’s angle. In the last step, the incidence angle θ was scanned with a recording time of 1 s for each step to obtain the reflectivity of the optics with respect to the incidence angle and, for the position of the intensity maximum, the profile of the reflected beam. Each reflectivity scan was repeated twice (5× for HAPG_126) and the mean was normalized to the incident flux of a measurement without optics taking into account the different absorber settings. The measured data were cropped and background-corrected by means of subtracting adjacent areas of the same dimension. Note that the recorded data themselves were dark frame corrected already, but the mean intensity drifted slightly with time.

### 3.2. Beam Homogeneity

Figure 2 shows the intensity distributions recorded with the“PE” detector for the direct beam, for each of the four aligned optics and the respective ray tracing simulations with expected mosaic spreads δ as given in Table 2. First of all, it can be seen that the overall position and shape of the measurements fit well with the expectations from the ray tracing simulation. The vertical width of the $1\times 1\;{\mathrm{mm}}^{2}$ direct beam is increased by about a factor of 10 to 20 for the different optics. Thus, the optics can be applied for a coarse scan in X-ray fluorescence imaging.

However, it is also apparent that the measured beam profiles are rather inhomogeneous, especially for the optics with applied HAPG crystals. The differences in intensities in each profile can be attributed to locally imperfect Bragg conditions, probably due to unexpectedly large tilts of single mosaic blocks. Indeed, the footprint size on the optics is only about $1\times 30\;{\mathrm{mm}}^{2}$, probably too little to guarantee a smooth angular distribution of the mosaic tilts. Additionally, adhesion might not be sufficient on the glass substrate, probably induced by the small radii of curvature at hand.

### 3.3. Reflectivity and Mosaic Spread

In Figure 3 the rocking curve results for each of the four optics are shown, as well as expected and adapted ray tracing simulations. The relative angular positions of the measurements are shifted to fit the reflectivity maximum to the Bragg’s angle of 2.00°.

For the ray tracing of the theoretical rocking curves, thicknesses provided by the vendor and mosaic spreads from Grigorieva et al. [11] are used, as well as the geometry of the beamline setup and optics dimensions. The rocking curves render full widths at half maximum of below 0.14° (for HAPG) and 0.44° (for HOPG), respectively, and peak reflectivities well above 17%.

The four measured rocking curves (blue dots) show much larger widths (>1.12° for HAPG and 0.62° for HOPG) and lower reflectivities (<3.1% for HAPG and 6.1% for HOPG) than expected (red dots) and, even more, the HAPG crystals render a worse reflection behavior than the HOPG optics. Additionally, especially in the case of HAPG_103 and HAPG_52, the shapes are rather asymmetric. For HOPG_129 and HAPG_52 a quick drop in reflectivity at about 3° can be seen and the reflectivity of HAPG_129 renders a sudden jump at 0.1°. Part of the asymmetry can be explained by geometric effects, namely the footprint of the incoming X-ray beam on the optics. Already for an incident angle of 2° the footprint on the optics is 30 mm in length, meaning that part of the beam is already missing the optics HAPG_103 and HAPG_52.

To address these effects, rocking curves by means of ray tracing with adapted crystal parameters (mosaic spread and mosaicity function) and an additional scaling factor *S* have been calculated. The used parameters are shown in Table 2. Note that while usually a Gaussian distribution describes the mosaicity function of HOPG material rather well [14], here we had to use a Lorentzian function to roughly describe the measured rocking curves.

As can be seen from the red curves in Figure 3, some of the asymmetries, especially for the shorter optics HAPG_103 and HAPG_52, can be explained with the adapted ray tracing. However, the spontaneous jumps in the reflectivity curves cannot be described by the simulation and indicate inhomogeneous crystallite distributions. Table 2 shows that the applied scaling factors are as low as 57% and drastic changes to the mosaic spreads have been applied. The intensity factors can result from two contributions. Either the thickness of the mosaic crystals is lower than assumed in the simulations (by a factor similar to the scaling factor), or only part of the crystal surface indeed contributes to the reflectivity, e.g., due to insufficient adhesion within the irradiated crystal area. While the former factor cannot be excluded, since the exact thickness of the HOPG and HAPG films is not known, it is rather unlikely that the values of the vendors are off at this scale. On the other hand, already the reflection profiles showed largely inhomogeneous intensity distributions, supporting that the second aspect is probably dominant. This would also explain the large widths of the rocking curves and spontaneous intensity jumps due to arbitrarily very strongly tilted crystallites. As a result, the expected peak reflectivities of up to 25% are not reached. Instead, HOPG_129 showed the best overall reflectivity with 6.1% and HAPG_129, with a measured peak reflectivity of 3.1%, has the highest value with respect to the HAPG optics. Note that the expected peak efficiency for HAPG_52 is reduced due to the small size of the optics.

Qualitatively, the results shown here were reproduced with a diffractometer equipped with a Cu anode X-ray tube in the vendors laboratory. The measured rocking curves show much larger widths than expected for the used crystals and large position-dependent inhomogeneities (Appendix A).

Since the HAPG optics were expected to show a better performance than the HOPG optics, HAPG_129 was chosen for the XFI measurements.

### 3.4. X-ray Fluorescence Imaging

As proof-of-principle for the application of defocusing optics for X-ray fluorescence imaging, a mouse-shaped phantom containing palladium marker solutions with different concentrations was scanned. Concentrations of 0.1 mg/cm^3^ in the liver, 0.2 mg/cm^3^ in the kidneys and 0.5 mg/cm^3^ in the lungs were chosen.

In a first measurement, the phantom was scanned with a pixel size of 1×5 mm^2^ using the beam deflected by the HAPG_129 crystal with 2 s acquisition time per pixel. This scan was used to obtain a full-body transmission image as well as to find the region in which Pd fluorescence was emitted. A total area of 76×35 mm^2^ was scanned. In a second measurement, the optics was moved out of the beam and the phantom stage was positioned in the direct synchrotron beam. Using the x/y slits, the incident beam size was reduced to 0.4 × 0.4 mm^2^ and the region of interest (31.2×26.8 mm^2^, corresponding to 31% of the coarse scan area) determined in the first run was scanned again to obtain a high-resolution image of the fluorescence distribution using 1 s acquisition time per pixel. Switching between the two scan modes did not require any manual rearrangement or interlock breaking and was performed within minutes. For this, the XFI stage including the detector was translated and rotated based on the Bragg angle and distance to the defocusing optics to align the setup for the respective configuration. Transmission images were obtained using the“X-ray eye” and fluorescence spectra were recorded using the Amptek SDD. Pd fluorescence counts were extracted by fitting using a Gaussian peak model and a cubic background function. For details, see Körnig et al., 2022 [1]. The resulting transmission/fluorescence maps are shown in Figure 4.

The region of the filled organs is clearly identifiable in the coarse scan, allowing good definition of the region of interest for the fine scan, in which the individual organs are revealed.

The combined measurement duration for both scans was 180 min (32 min for the coarse scan and 148 min for the fine scan) compared to 471 min, which would be needed for a full scan at fine resolution. However, those numbers include the time needed for the small motor movements on step-wise scans, which take around 40% of the total measurement time but can be strongly reduced using a continuous scan [1]. The raw detector’s acquisition times without motor movements are 18 min for the coarse and 87 min for the fine scan and around 280 min for a full fine scan. Either way, total measurement time was reduced by more than 60% when using the combined scan approach.

As compared to the full fine scan, a reduction in dose uptake has been achieved. In first order, this can be estimated by weighting the measurement time of the coarse scan with the measured peak reflectivity of 3.1% of the defocusing optics and comparing the combined (weighted) scan time of coarse plus fine scan to the scan time of the full fine scan. The result shows a reduction in dose of 68%. However, it has to be noted that the fine scan uses a factor of 2.7 more incident photons per pixel than the coarse scan in this proof-of-principle measurement, resulting in a higher sensitivity. This means that a potential region of interest might not be found in the coarse scan but would have been identified in a full fine scan. In an optimized scenario, the same number of photons for pixels of each scan should be used to have the same sensitivity regarding fluorescence marker mass per area in both datasets. Then, the overall dose uptake of the sample by the coarse scan as compared to the full fine scan would be lower by the factor of increased beam size, which most often will lead to reduced overall dose uptake.

Clearly, the actual numbers will differ for each sample depending on the size of the actual region of interest as well as choice of necessary statistics in each scan, but the suggested strategy allows for more flexibility and measurement schemes tailored to the sample requirements. This is especially important with respect to time and dose restrictions for in vivo investigations.

## 4. Conclusions

It has been shown that by means of cylindrically shaped reflection optics based on pyrolytic graphite mosaic crystals a parallel, 1×1 mm^2^ sized synchrotron beam could be enlarged vertically by a factor of 10 to 20 (depending on the optics radius) within a distance of 1.5 m. The measured efficiency is 1.6% to 6.1% for the different optics and thus does not match the expectations of more than 17%. The investigation of the reflection profiles and the comparison to ray tracing simulations and similar crystals on flat substrates leads to the conclusion that the overall reflection properties (reflectivity, homogeneity) are not ideal, yet. However, since the optics presented here are the first optics with convex shape manufactured by the vendor, further enhancements by adapted application processes for the graphite thin films can be expected.

Using optics HAPG_129, proof-of-principle XFI measurements on a mouse phantom demonstrated the benefit and usability of the defocusing concept. Even though the fluorescence marker was distributed in a large fraction of the total size of the mouse, the applied coarse scan with a 5× enlarged synchrotron beam allowed to strongly reduce the overall dose (by 68%) and measurement time (by 62%) of the investigation. However, the dose and time reduction strongly depend on the localization of the fluorescence marker. In applications with more localized distributions—and therefore a smaller region of interest—even larger reductions can be expected while in the (unrealistic) case of a homogeneous distribution, only a slight increase (≈3%) of both can occur. Additionally, future enhancements of the optics reflection properties will further reduce measurement times of the coarse scan. While the demonstrated measurement time of 180 min is still too long for in vivo investigations, the experimental setup was designed to only demonstrate the relative dose and time reduction using defocusing optics. Possible methods to reduce the measurement duration are the switch from a step-based to a continuous scan and an improved detector placement closer to the sample. Furthermore, acquisition times per pixel have not been optimized to the minimal required duration for a statistically significant XFI signal and can be reduced in the future. By implementing these improvements, the total acquisition time can be reduced to in vivo conform levels.

Finally, we want to note that the findings described here are by no means limited to X-ray fluorescence imaging. Other synchrotron-based techniques with requirements for flexible and larger beam sizes, directly controllable by the user, e.g., in coherent diffraction imaging or X-ray scattering, might also benefit from the presented results.

## Figures and Tables

**Figure 1 ijms-23-04673-f001:**
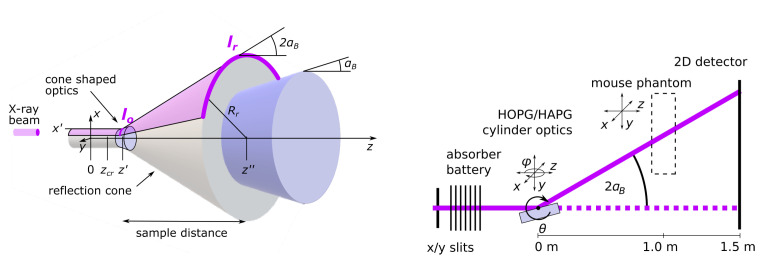
(**Left**): Schematic view of diverging optics. (**Right**): Schematic view of the experimental setup.

**Figure 2 ijms-23-04673-f002:**
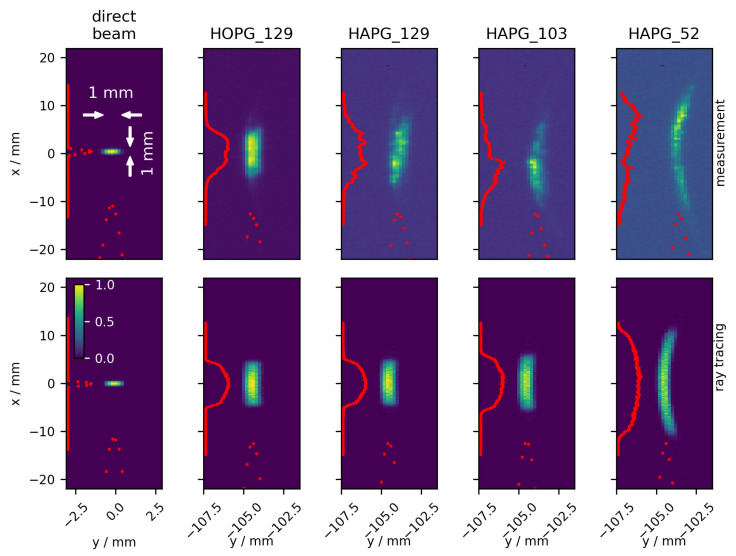
Measured (**top**, “PE” detector) and simulated (**bottom**) beam intensities at a distance of 1.5 m from the optics position. The left-most column shows the direct synchrotron beam without reflection optics. The other columns refer to the beam profiles of the respective optics when the reflectance is maximum within the rocking curve. While the positions are given in absolute values, the intensity is normalized for reasons of clarity. The general shape and position of beam profiles meet the prediction of the ray tracing simulation. The measured inhomogeneities are likely due to imperfect crystal adhesion (see text for details).

**Figure 3 ijms-23-04673-f003:**
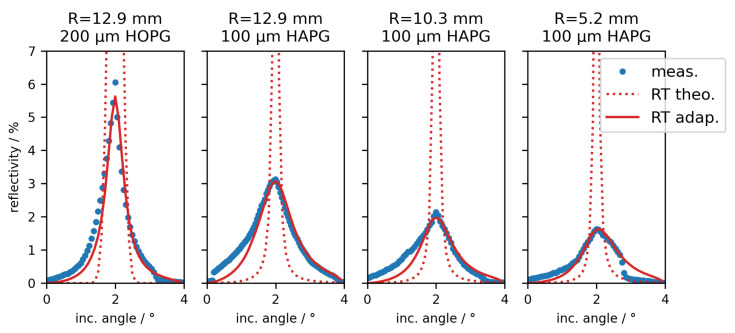
Measured (blue dots) and simulated (red dots and line) rocking curves for the four diverging optics. The dotted red line depicts the expected reflection behavior using the crystal parameters given in Table 2 by means of ray tracing simulations. The solid line corresponds to ray tracing simulations using the adapted parameters given in the same table.

**Figure 4 ijms-23-04673-f004:**
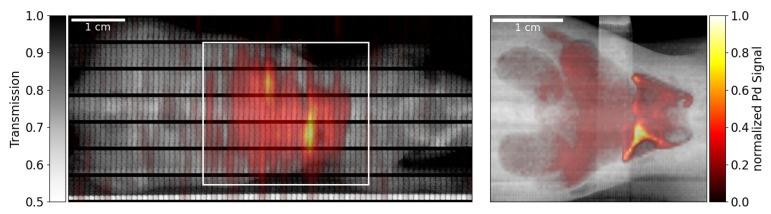
Composite transmission/fluorescence images for the coarse scan using the diverging optics (**left**) and the fine scan of the signal region using the direct beam (**right**). The horizontal black bars in the coarse transmission image are a result of the limited field of view of the “X-ray eye”. The fluorescence map is normalized to its maximum value for each individual scan.

**Table 1 ijms-23-04673-t001:** Properties of the applied diverging optics.

Name	Substrate id	Cylinder Radius /mm	w × l /mm	Crystal Type
HOPG_129	LJ1075L2	12.9	20 × 40	HOPG
HAPG_129	LJ1075L2	12.9	20 × 40	HAPG
HAPG_103	LJ1328L2	10.3	15 × 30	HAPG
HAPG_52	LJ1878L2	5.2	10 × 20	HAPG

**Table 2 ijms-23-04673-t002:** Adapted crystal parameters used for the ray tracing simulation. Expected thickness values were given by the vendor (Optigraph GmbH) and expected mosaic spreads δ are taken from Grigorieva et al. [11]. These values lead to the expected peak reflectivity values $R_{\mathrm{theo}}$. The expected mosaicity function for HOPG is Gaussian and for HAPG Lorentzian. However, to adapt the ray tracing to the data also for the HOPG crystal a Lorentzian mosaicity function had to be applied. Additionally, a scaling factor *S* had to be used to account for deviating thicknesses, insufficient crystal adhesion and uncertainties in tabulated integral reflectivities. Thus, the measured (and adapted) peak reflectivity $R_{\mathrm{meas}}$ is lower than expected.

Name	Thickness /μm	δ/° Expected	$R_{\mathrm{theo}}$	δ/° Adapted	*S*	$R_{\mathrm{meas}}$
HOPG_129	200	0.4	17.9%	0.56	57%	6.1%
HAPG_129	100	0.1	24.9%	1.18	109%	3.1%
HAPG_103	100	0.1	24.9%	1.16	70%	2.1%
HAPG_52	100	0.1	19.6%	1.21	74%	1.6%

## Data Availability

Not applicable.

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
