# Peer review of "Enabling Coarse X-ray Fluorescence Imaging Scans with Enlarged Synchrotron Beam by Means of Mosaic Crystal Defocusing Optics"

_ijms, 2022, doi:10.3390/ijms23094673_

Round 1
Reviewer 1 Report
The authors present the concept of graphite based, convex reflection optics for one-dimensional enlargement of a 1 mm wide synchrotron beam by a factor of 5 to 10 within 1 m distance. Four different optics are tested and characterized and their reflection properties compared to ray tracing simulations. The general shape and size of the measured reflection profiles agree with expectations. Enhancements with respect to homogeneity and efficiency can be expected with improved optics manufacturing. Overall, the article is well organized and its presentation is good and can be accepted in the present form.
Author Response
Thank you for your positive feedback. We changed a few parts of the manuscript to clarify some remarks of Reviewer 2.
Reviewer 2 Report
This paper is clearly written with a high intellectual honesty. The paper is of interest for colleagues that work in the field of X-ray fluorescence imaging. Although, it is focused, the paper lacks comparisons in imaging performance (acquisition time, spatial resolution, sensitivity) with other classic whole body fluorescence imaging technique. I recommend publication after minor revision.
Minor remarks:
Abstract: line 1: functionalized nanoparticles and labeled entities can be localized with sub mm spatial resolution by X-ray fluorescence imaging (XFI). Similar spatial resolutions are obtained with classic whole body fluorescence imaging and this technique is much faster? What is the advantage of X-ray fluorescence, you can excite specific ‘spore elements’ at low concentration, but the diffusion of light photons (fluorescence) is limited in tissues. What is the imaging depth of XFI ?
Introduction : line 31, “synchrotrons are needed to minimize background contributions in the fluorescence energy region”. This is not true, imaging and microscopy techniques with infrared excitation wavelengths limit background contributions. There is no background in XFI ?
Results and discussion, paragraph 3.3 line 236 : Since the HAPG optics were expected to show a better performance than the HOPG 236 optics, HAPG_129 was chosen for the XFI measurements.
Ok this is expected but this is not measured? In figure 3, the measured reflectivity (%) was higher for HOPG 236 optics in comparison to HAPG-129 optics? Why is HOPG 236 optics not chosen for the XFI measurements?
Line 263 – 265: The combined measurement duration for both scans was 180 minutes - 32 minutes for the coarse scan and 148 minutes for the fine scan - compared to 471 minutes, which would be needed for a full scan at fine resolution.
The acquisition time are too long for biomedical imaging applications. Discus here, options to reduce the acquisition time.
